# Does technological innovation in National Sustainable Development Agenda Innovation Demonstration Zones promote green development?—the case from Chengde City, China

**Qingqing Yuan[1], Guofeng Zhang[1,2,3]*, Ruixian Wang[1], Xiaojing Ma[4], Jiangao Niu[1,2]**

**1** School of Economics, Hebei GEO University, Shijiazhuang, 050031, China, **2** Research Base for Scientific-Technological Innovation and Regional Economic Sustainable Development of Hebei Province, Hebei GEO University, Shijiazhuang, 050031, China, **3** Hebei Province Mineral Resources Development and Management and the Transformation and Upgrading of Resources Industry Soft Science Research Base, Shijiazhuang, 050031, China, **4** School of Earth Sciences, Hebei GEO University, Shijiazhuang, 050031, China

* zhangguofeng@hgu.edu.cn

**Data Availability Statement:** This information will only be available after acceptance.

**Funding:** Funding for this paper was provided by the Science and Technology Planning Project of

## Abstract

The National Sustainable Development Agenda Innovation Demonstration Zones (NSDAIDZs) aim to spearhead green development through scientific and technological innovation, showcasing sustainable development to other regions in China and offering valuable insights for countries worldwide. Taking Chengde City, which is one of the cities in the second batch of NSDAIDZs, as a case study, we examine the quantitative impact of technological innovation on green development. Additionally, it investigates the threshold effect of Research and development investments (R&D investments) on the relationship between technological innovation and green development. The results indicate that: (1) technological innovation has a positive promoting effect on green development, with a 1.01% increase in green development for every one unit increase in technological innovation; (2) The positive effect of technological innovation on green development becomes fully realized only when R&D investments and the upgrading of industrial structure surpass a specific threshold value. We contribute to the existing research on the connection between technological innovation and green development in innovation demonstration zones. It also provides empirical insights to foster a mutually beneficial relationship between R&D investments, industrial structure upgrading, and technological innovation, ultimately maximizing the promoting role of technological innovation in green development.

## Introduction

The definition of sustainable development has been broadened from "meeting the needs of the present without compromising the ability of future generations to meet their own needs" to

Hebei Provincial Department of Science and Technology (decision number: 225576117D) awarded to GZ, the Key Program of Humanities and Social Sciences of Hebei Provincial Department of Education (decision number: ZD202311) awarded to JN, and the Graduate Student Innovation Ability Training Funding Project of Hebei Province (decision number: CXZZSS2023137) awarded to QY. The funders had no role in study design, data collection and analysis, decision to publish, or preparation of the manuscript. The authors gratefully acknowledge the support of the above projects.

**Competing interests:** The authors have declared that no competing interests exist.

"inclusive well-being"–the aggregate quality of life for all people, everywhere, now and in the future-does not decline with time" [1,2]. In September 2015, the United Nations Development Summit adopted the 2030 Agenda for Sustainable Development, which established global Sustainable Development Goals (SDGs) that provide specific targets for enhancing inclusive well-being [3].

A number of organizations and countries have incorporated the 2030 Agenda into their national development plans [4,5]. At the same time, scholars worldwide have conducted extensive research on the implementation of SDGs, particularly on SDG 3 [6,7], SDG 4 [8,9], SDG 6 [10], and SDG 12 [11,12]. China, along with the international community, is actively advancing the implementation of a global sustainable development strategy (Fig 1). Since 2016, the State Council of China initiated the establishment of 11 National Sustainable Development Agenda Innovation Demonstration Zones (NSDAIDZs) (Fig 2). Each demonstration area has its specific implementation content and characteristics.

Technological innovation has strong social effect, and is also the key to enhance global sustainable development [13]. However, can technological innovation prove effective in actualizing green development in National Sustainable Development Agenda Innovation Demonstration Zones? In addition, does its impact differ under different conditions of R&D investments and industrial structure? It is of great theoretical and practical significance to explore and explain these issues in depth. Delving into and elucidating these inquiries holds immense theoretical and practical importance. This exploration not only unlocks the dynamic potential of technological innovation within the framework of the National Sustainable Development Agenda Innovation Demonstration Zones, making a substantial contribution to eco-friendly development, but also offers valuable experience and reference for other countries internationally grappling with similar sustainable development challenges.

It is widely recognized that technological innovation plays a pivotal role in promoting industrial development and fostering economic growth. Based on a dataset of 85 regions in Russia, Aldieri [14] found that technological innovation has a significant positive impact on regional economic growth and transformation and upgrading of industrial structure. He [15] analyzed data from 38 Asian countries using statistical models like unit root tests, the Westerlund cointegration test, and AMG regression models. Their study emphasized the pivotal role of technological innovation in achieving sustainable economic growth. Moreover, some scholars have suggested that the impact of technological innovation on economic growth extends through spillover effects. For instance, Org [16] proposed in the context of new economic geography that technological innovation and its spatial diffusion result in knowledge spillover effects. These effects, in turn, accelerate regional industrial transformation and upgrading while fostering sustainable economic growth.

The relationship between technological innovation and green development is a subject of ongoing debate among scholars, and consensus has not been reached. Some scholars believe that technological innovation plays a driving role in green development. It can improve factor resource productivity [17], reduce pollutant emissions [18], enhance waste treatment [19], enable more effective secondary recycling of resources [20], reduce the output of heavily polluting industries [21] and reduce carbon emissions [22]. This is achieved by accelerating technology transfer and transformation, where the impact of green technological innovation is more obvious [17]. For instance, Zhang [23] suggested that green technological innovation can facilitate emerging economies' transition from high pollution to a stage of sustainable development, as indicated by the Environmental Kuznets Curve.

However, it has also been argued that technological innovation can be perceived to hinder green development. The opportunity cost effect argues that technological innovation has an opportunity cost. Meanwhile, the cash flow effect argues that technological innovation is risky

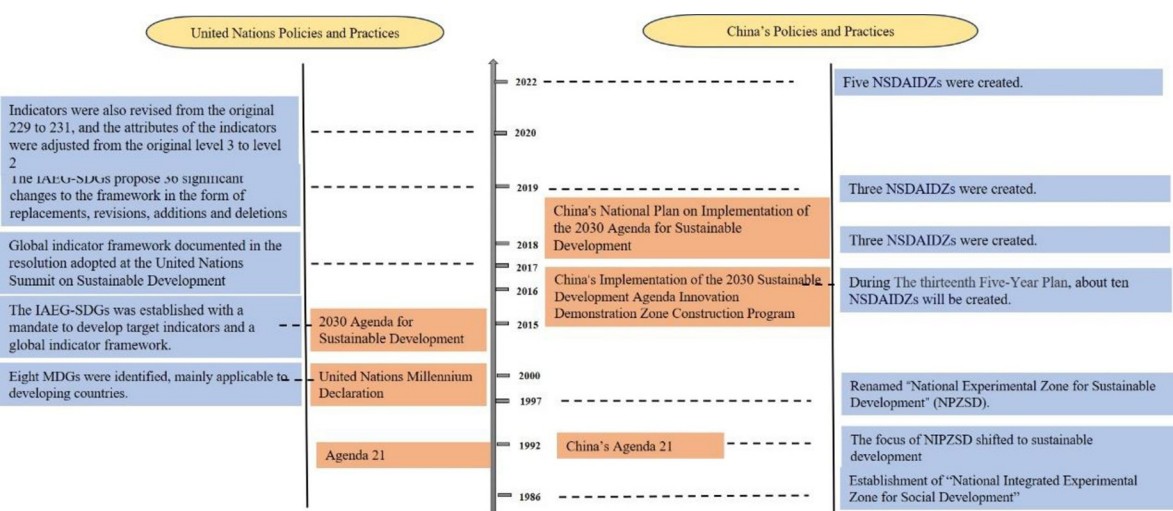

**Fig 1. The Sustainable development process in China and internationally.**

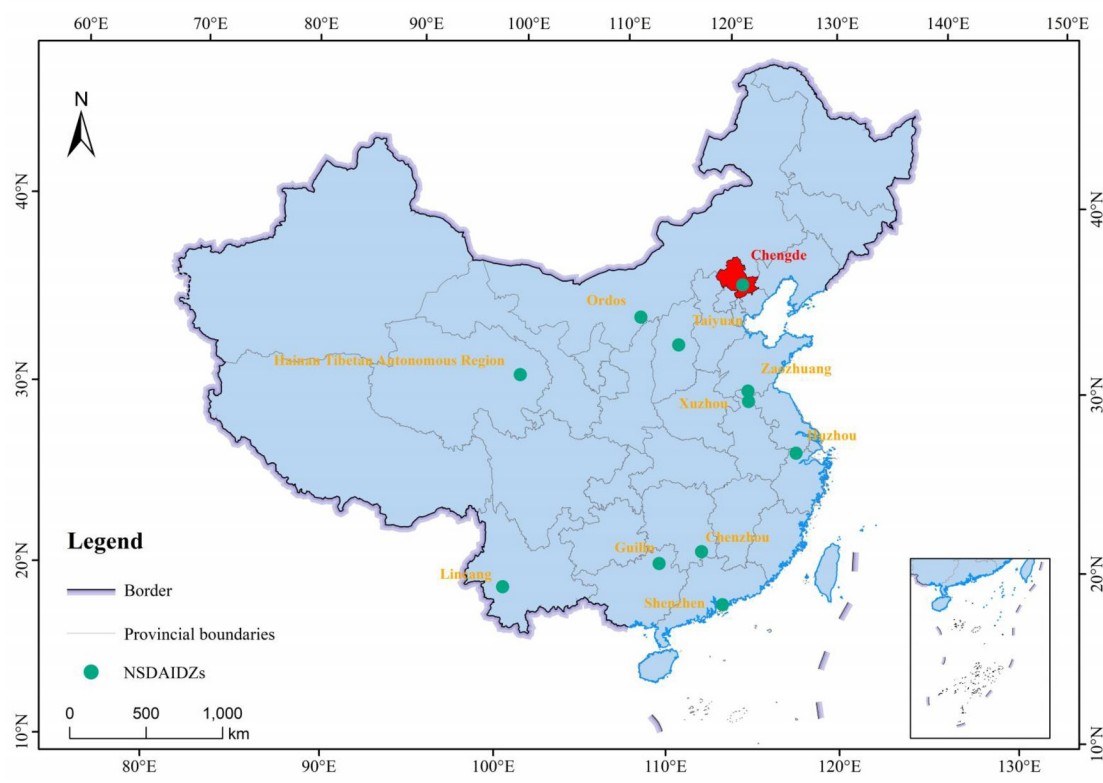

**Note: This map is derived from the standard map available on the website of the Ministry of Natural Resources of the People's Republic of China [Map Review Number: No. GS (2023) 2767, URL: http://www.mnr.gov.cn].**

**Fig 2. The overview of the distribution of National Sustainable Development Agenda Innovation Demonstration Zones.**

and uncertain. When the rate of technology transfer is low, the benefits of technological innovation may not outweigh its opportunity cost [24]. Moreover, technological innovation can trigger a "rebound effect". For instance, there is the possibility of an "energy rebound effect" where profit-driven enterprises, in their pursuit of cost savings and labor efficiency during technological innovation, might disregard potential ecological harm. This oversight can result in increased energy consumption and pollutant emissions during the product manufacturing process.

Some other scholars believe that a non-linear relationship exists between technological innovation and green development. At lower levels of technological innovation, resource and energy utilization efficiency tends to be low. However, with the advancement of technological innovation, the widespread adoption of advanced and cleaner production technologies occurs, facilitating a reduction in resource and energy consumption as well as emissions of pollutants. This dynamic gives rise to the formation of a "U"-shaped curve in the relationship between technological innovation and green development [25].

The relationship between technological innovation, industrial structure, R&D investment and green development is also an area of continuing scholarly interest. Song [24] argued that the shift to green technology-based production is imperative for achieving sustainable development, especially in key industries. The efficiency of technological innovation, and the advanced industrial structure are all conducive to the improvement of green development in manufacturing industry [26]. If there is an increase in the spatial imbalance of green technological innovation, industrial structure rationalization, and industrial structure advancement, the gap between carbon emission efficiency will be increased, and regional sustainable development will be adversely affected [27]. Meanwhile, R&D investment is an important driving force for regional green innovation [28]. In recent years, China has also actively implemented the innovation-driven development strategy and has continuously increased its R&D investment [29]. While R&D investment has continued to increase, China's environmental pollution has become increasingly serious [30]. As a result, R&D investment and green innovation have failed to harmonize. Therefore, the issue of how to rationally allocate R&D resources so as to improve the regional green innovation performance effectively has become increasingly critical. In recent years, the development of digital technology has given rise to a digital finance model, which has effectively improved the allocation of capital factors [31]. In addition, under the policy support for sustainable development, financial institutions are likely to provide various resources to polluting firms to support their green development [32].

To sum up, the existing research provides a solid and comprehensive theoretical foundation for studying the relationship between technological innovation and green development. However, there are still areas for expansion. First, there remains a lack of consensus among scholars regarding whether technological innovation effectively promotes green development, and research on the NSDAIDZs has been neglected. Second, the existing research mainly focuses on the relationship between R&D investments, industrial structure, technological innovation, and green development, often failing to integrate these factors within a unified research framework. This ignores the significant roles of R&D investments, industrial structure, technological innovation and green development.

Given this, we supplement and expand in the following ways: first, we examine the relationship between technological innovation and green development using a panel fixed-effects model and verify the reliability of the regression results through a series of robustness tests. Second, R&D investments and industrial structure upgrading are introduced as threshold variables into the analytical framework of the relationship between technological innovation and green development. The impact of technological innovation on green development under different levels of R&D investments and industrial structure upgrading is discussed from the perspective of dynamic regulation.

Therefore, the contribution of this paper is as follows: First, a set of comprehensive evaluation model is constructed to analyze the impact of technological innovation in the NSDAIDZs on green development based on the background of the sustainable development. Second, complete evaluation of the effect of technological innovation on green development before and after the policy of the establishment of NSDAIDZs. Third, Integrating the factors of R&D investment, industrial structure and technological innovation into a unified research framework, this paper analyzes whether R&D investment, industrial structure and technological innovation will form the ability to coordinate development and promote green development.

## Materials and methods

### Mechanism and research hypotheses

Impact of technological innovation on green development. The demonstration zone deeply implements the innovation-driven development strategy, focuses on the high-quality development of "3+3" leading industries, strengthens high-tech research and the transformation of achievements, and promotes the gathering of talents, technology, capital and other elements to the demonstration zone. Technological innovation serves as the core driving force and vital support for green development. It enhances resource utilization efficiency and facilitates the transformation of cities towards a circular economic model. Simultaneously, enterprises, through technological innovation, accelerate the research development and promotion of clean energy such as wind energy, hydropower, nuclear power, wind power, solar power, and other renewable energies, reducing reliance on natural resources. In addition, technological innovation can improve the level of production intelligence by strengthening the technological iteration of industrial production processes, thus fostering the green development of cities. The importance of technological innovation in green development has been confirmed. However, scholars continue to debate whether the relationship between technological innovation and green development is linear. It is not universally true that higher levels of technological innovation always result in more significant promotion of green development. When the level of technological innovation is low, the efficiency of resource and energy utilization is low; along with the enhancement of technological innovation, advanced cleaner and other production technologies have been widely used, promoting the reduction of resource and energy consumption and emissions of three wastes. Therefore, there may be a U-shaped relationship between technological innovation and green development. In light of these considerations analysis, we propose hypothesis H1.

H1: Technological innovation has a significant positive effect on green development, but there is no U-shaped relationship between the two.

Nonlinear effect of R&D investments on the impact of technological innovation on green development. R&D investments serves as a crucial foundation for technological innovation, yet it also constrains its progress. The extent of R&D investments directly shapes countries' or regions' innovative outcomes. For the ensuing two reasons, a potential threshold effect of R&D investments on the impact of technological innovation on green development may arise.

First, in cases where R&D investment falls short, the anticipated benefits may remain unrealized. In terms of innovation theory interpretation, it is acknowledged that innovation activities necessitate substantial R&D investment. However, a simplistic linear relationship does not align with the fundamental attributes of R&D endeavors. Throughout the innovation process, a significant infusion of talent, capital, and other R&D resources occurs in the early stages. It is only through persistent accumulation that the level of technological innovation can be enhanced.

Second, purely relying on direct internal R&D investments may lead to a diminishing impact on promoting green development over time. Excessive focus on the quantity of R&D investments may lead to monitoring of the efficiency and rationality of resource allocation. When R & D investment is too much, it will also lead to waste of resources. Therefore, the impact of technological innovation on green development is contingent upon the magnitude of R&D investment, exhibiting a discernible threshold effect. This paper utilizes GD as the green development index, TI as the innovation output index, and RD as the R&D investment element. The specific form of the breakpoint model is defined as follows:

$$
\begin{cases}
GD = \alpha_1 TI + \varepsilon, RD < \ Threshold_1 \\
GD = \alpha_2 TI + \varepsilon, \ Threshold_1 \leq RD \leq \ Threshold_2 \\
GD = \alpha_3 \ TI + \varepsilon, RD > \ Threshold_2
\end{cases}
\tag{1}
$$

When R&D investment is insufficient or excessive, the effect of technological innovation on green development will be weakened, i.e. $\alpha_1 < \alpha_2, \alpha_3 < \alpha_2$. Therefore, only when R&D investments and technological innovation form a benign interactive relationship, can the promotion effect of both on green development be maximized. Thus, we propose hypothesis H2.

H2: R&D investments as a threshold variable make the impact of technological innovation on green development have "non-linear characteristics".

**Non-linear effect of industrial structure upgrading on the impact of technological innovation on green development.**   The pivotal factor in assessing industrial structure upgrading lies in evaluating the extent of interaction among various industries, the rationality of the industry composition ratios, and the coordination of their development speeds. The upgrading of the industrial structure involves optimizing resources allocation and improving the efficiency of the industrial structure by adjusting the balance among different industries. The non-linear nature of the impact of technological innovation on green development may also arise from the threshold condition of industrial structure upgrading.

First, industrial structure upgrading positively moderates the impact of technological innovation on green development. The core mechanism of industrial structure upgrading to enhance green development lies in its ability to promote increased output levels, save factor inputs, and enhance the efficiency of green development by generating new green technologies and creating new market demand. The influence of technological innovation on the efficiency of green development increases with the level of industrial structure upgrading. This paper utilizes GD as the green development index, TI as the innovation output index, and STR as the industrial structure upgrading element. The specific form of the breakpoint model is defined as follows:

$$
\begin{cases}
GD = \beta_1 TI + \varepsilon, STR < \ Threshold \\
GD = \beta_2 TI + \varepsilon, STR \geq \ Threshold
\end{cases}
\tag{2}
$$

Second, the impact of industrial structure upgrading on green development exhibits non-linear characteristics. Different levels of industrial structure upgrading can result in heterogeneity in the relationship between technological innovation and green development. As industrial structure upgrading reaches a certain level, the shift from a focus on secondary industry to tertiary industry becomes prominent. This transition is expected to lead to a reduction in pollutant emissions, amplify the "green effect," and stimulate green development. A higher level of industrial structure upgrading means that the local area possesses a more robust

foundation for the development of green industry. In such circumstances, technological innovation can play a better role in promoting the construction of green and low-carbon production systems to support the green development of the local area. Thus, we propose hypothesis H3.

H3: Industrial structure upgrading as a threshold variable makes the impact of technological innovation on green development have "non-linear characteristics".

The mechanism framework is displayed in Fig 3.

## Research models

To reveal the impact of technological innovation on green development, we construct the following benchmark econometric model:

$$GD_{it} = \alpha_0 + \alpha_1 TI_{it} + \alpha_c\, Control_{it} + \mu_i + \delta_t + \varepsilon_{it} \tag{3}$$

$$GD_{it} = \beta_0 + \beta_1 TI_{it} + \beta_2 TI_{it}^2 + \beta_c\, Control_{it} + \mu_i + \delta_t + \varepsilon_{it} \tag{4}$$

Where the subscripts $i$ and $t$ denote the region and year respectively; $GD$ denotes the explanatory variable (green development); $TI$ denotes the core explanatory variable (technological innovation); $Control$ denotes a series of control variables, and $\mu_i$ and $\delta_t$ denote the individual and time fixed effects respectively, and $\varepsilon_{it}$ denotes the random error term. Eq (3) is used to test the impact of technological innovation on green development, and Eq (4) adds the quadratic term of technological innovation, which proves that there is a significant inverted U-shaped relationship between technological innovation and green development if the regression coefficients of the quadratic term are significantly negative and the regression coefficients of the primary term are significantly positive.

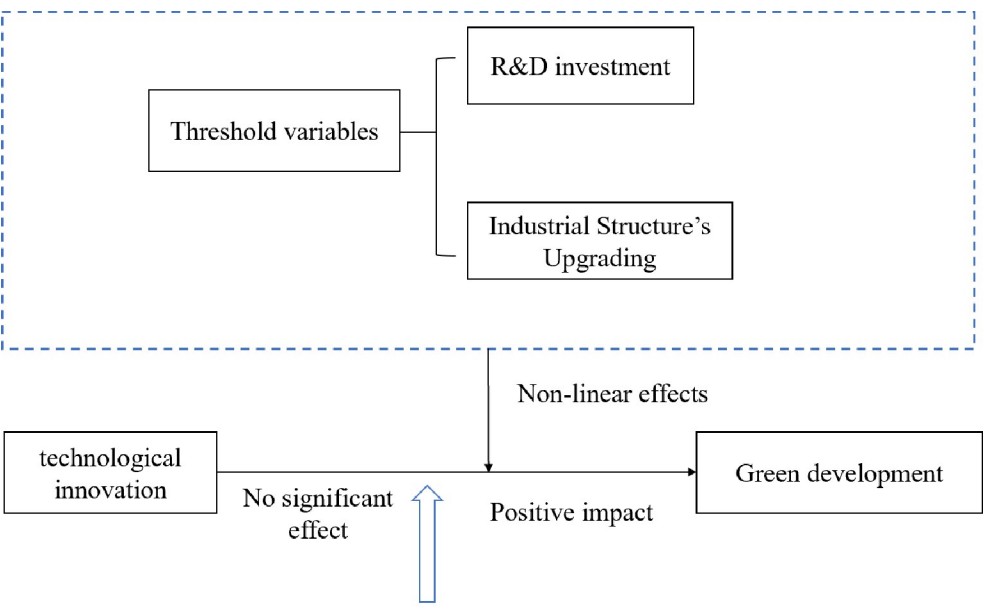

**Fig 3. Research framework of the impact of technological innovation in National Sustainable Development Agenda Innovation Demonstration Zones on green development.**

To test the dynamic nonlinear relationship between technological innovation and green development, we draw on Hansen's (1998) threshold model construction method. In this model, technological innovation serves as the core explanatory variable affected by the threshold variables, namely R&D investments and industrial structure upgrading. We employ green development as the explanatory variable to construct the panel threshold regression model. Since it is initially challenging to ascertain the number and estimated values of thresholds, we begin with a triple threshold model. Other types of threshold models can be derived from this base model. The specific form of the threshold model is set as follows:

$$GD_{it} = \beta_0 + \beta_1 TI_{it} \times I(Adj_{it} \leq \gamma_1) + \beta_2 TI \times I(\gamma_1 < Adj_{it} \leq \gamma_2) + \beta_3 TI_{it} \times$$
$$I(\gamma_2 < Adj_{it} \leq \gamma_3) + \beta_4 TI_{it} \times I(Adj_{it} > \gamma_3) + \beta_c\, Control_{it} + \mu_i + \delta_t + \varepsilon_{it} \tag{5}$$

where $Adj$ is the threshold variable, $\gamma_1$, $\gamma_2$, and $\gamma_3$ are the triple thresholds to be estimated, and $I(\cdot)$ represents a schematic function that takes on a value of 1 when the conditions within the parentheses are met, and 0 when they are not. The remaining symbols adhere to the conventions outlined in Eq (4).

## Variable selection

Explained variable: Green development level (GD). We adopt the entropy method to measure the level of green development. In 2016, the National Development and Reform Commission published the Green Development Indicator System, which comprehensively reflects the connotation and essence of the new concept of green development. However, due to the availability of data at the district and county level, we appropriately adjust the original indicators to construct the green development indicator evaluation system of Chengde. Specifically, refer to Table 1. The green development level index for 11 counties (cities and districts) in Chengde City is measured using the entropy method. The weight coefficients of environmental governance, growth quality, and ecological protection are determined as 0.160, 0.431, and 0.409, respectively.

Further, we apply the kernel density estimation method to analyze the dynamic evolution law of the distribution of the green development level of 11 counties (cities and districts) in Chengde City, as shown in Fig 4. As can be seen from Fig 4, the overall kernel density curve of the green development level of 11 counties in Chengde City has exhibited a rightward (larger value) shift from 2014 to 2020. This indicates a noticeable upward trend in the level of green

**Table 1. Green development indicator system.**

| First-level indicators | Second-level indicators | Unit | Attribute | Weight |
|---|---|---|---|---|
| Environmental governance | Harmless treatment rate of domestic garbage | % | Positive | 0.073 |
| | Centralized sewage treatment rate | % | Positive | 0.087 |
| Quality of growth | growth rate of GDP per capita | % | Positive | 0.013 |
| | Disposable income of urban residents | CNY | Positive | 0.042 |
| | Disposable income of rural residents | CNY | Positive | 0.042 |
| | Value-added of tertiary industry as a share of GDP | % | Positive | 0.057 |
| | Expenditure on research and experimental development as a share of GDP | % | Positive | 0.255 |
| Ecological protection | Annual average PM2.5 concentration | $\mu g/m_3$ | Negative | 0.048 |
| | Proportion of days with air quality reaching Class II | % | Positive | 0.013 |
| | Fertilizer use per unit of sown area of crops | kg/ha | Negative | 0.159 |
| | Pesticide use per unit of sown area of crops | kg/ha | Negative | 0.211 |

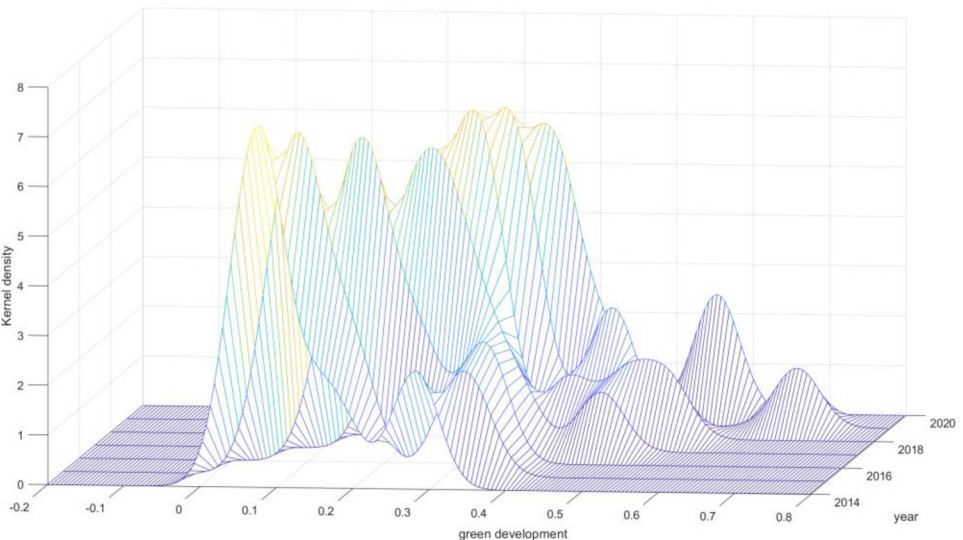

**Fig 4. Kernel density map of green development level.**

development in Chengde City in recent years. The center of the kernel density function, along with the range of change, has shifted consistently to the right, indicating a clear increase in the level. The distribution maintains a single-peak pattern, without any signs of polarization. The height of the primary peak initially increases, followed by a decrease. The higher the "peak," the more "dense" the data. Therefore, counties with high and low levels of green development are both close to the average level. The width of the curve remains relatively consistent, indicating that the width of the curve is generally not very different, indicating that the absolute difference in green development level among the 11 counties in Chengde tends to be narrowed.

Core explanatory variable: Technological Innovation (TI). Currently, there is a large amount of relevant literature involving the measurement of technological innovation. The established literature generally uses patent data to measure the level of technological innovation. Patent types include invention patents, utility model patents, and design patents, of which invention patents are more novel than the other two patent types [33] and belong to higher-quality innovation. The effective transformation and utilization of invention patents can bring about shifts in demand structure and labor productivity through technological breakthroughs and enhancements. This, in turn, is advantageous for stimulating the enthusiasm of various factors of production and enhancing overall production efficiency [30]. In addition, patent data also include patent applications and patent grants, where the number of patent applications is more time-sensitive and less likely to be affected by the time lag. At the same time, to eliminate the impact of population size, we choose the number of invention patent applications per 10,000 people to measure the level of technological innovation.

Threshold Variable: Research and development investment (RD). At present, the measurement of R&D investments is mainly divided into two forms. One is to directly use the number of R&D personnel and the number of R&D investments to measure [34]. The other is to use the number of R&D investments as a proportion of operating income indicator to measure [35]. We adopt the latter to measure the level of R&D investments.

Industrial structure upgrading (STR). Currently, there is a substantial body of literature involving the measurement of industrial structure upgrading [36,37]. Typically, we measure industrial structure upgrading by considering two key dimensions: the advancement of

industrial structure and the rationalization of industrial structure. Due to the limitation of the availability of county data, we measure the level coefficients of industrial structure, which is obtained by multiplying the proportion of output value of the primary, secondary, and tertiary industries in the GDP by 1, 2, and 3. The higher the level coefficient of industrial structure, the greater the transfer of industry from low to high, leading to a more pronounced enhancement in industrial structure.

Control variables. Four indicators were selected to best reflect the impact of urbanization level, population, and finance on green development. The four control variables: (1) urbanization level (UR): calculated as the ratio of urban population to total population; (2) fiscal expenditure scale (FE): defined as the ratio of local fiscal expenditure to GDP; (3) population density (PD): calculated as the ratio of year-end resident population to the area of administrative area; (4) fiscal revenue scale (FR): defined as the ratio of local fiscal revenue to GDP; To eliminate the difference in the scale, the variable population density is logarithmized.

## Data source

The statistical description of the data is specified in Table 2, where missing indicators for some years are treated using linear interpolation and mean replacement. To eliminate the effect of inflation, all economic and social data are adjusted according to the 2014 GDP deflator. Limited to the availability and completeness of county data, we select the panel data of 11 counties (cities and districts) in Chengde City from 2014 to 2020, and the data are mainly derived from the Hebei Economic Yearbook (changed to Hebei Statistical Yearbook in 2020 and later), Chengde Statistical Yearbook, Chengde Ecological Environment Condition Bulletin, and China Urban Construction Statistical Yearbook. In addition, the data on the number of patent applications received were obtained from the State Intellectual Property Office (SIPO), and the data on PM2.5 concentrations were obtained from the Atmospheric Composition Analysis Group (ACAG) [38].

## Results and discussion

Based on the theoretical analytical framework of technological innovation affecting green development described in the previous section, we test the mechanism of the role of technological innovation in promoting green development from the following two aspects: (i) the benchmark regression model and the endogeneity and robustness test are used to verify the role of technological innovation in promoting green development; (ii) the panel threshold effect model is used to test whether there is a threshold effect of R&D investments in technological innovation on green development.

**Table 2. Descriptive statistics of variables.**

| Variables | Sample | Size | Mean Value | Standard Deviation | Minimum Value | Maximum Value |
|---|---|---|---|---|---|---|
| Green Development | GD | 77 | 0.2580 | 0.1270 | 0.0460 | 0.6780 |
| R&D investments | RD | 77 | 0.6850 | 0.8500 | 0.0030 | 3.869 |
| Technological Innovation | TI | 77 | 0.9990 | 1.8560 | 0 | 8.8110 |
| Industrial Structure's Upgrading | STR | 77 | 2.1970 | 0.1890 | 1.8900 | 2.7900 |
| Urbanization Level | UR | 77 | 54.4580 | 20.9030 | 32.0000 | 95.5300 |
| Scale of Fiscal Expenditure | FE | 77 | 20.4550 | 9.0180 | 6.3550 | 47.7350 |
| Population Density | PD | 77 | 4.9330 | 0.8540 | 3.7210 | 6.5670 |
| Fiscal Revenue Size | FR | 77 | 5.5460 | 2.0620 | 2.5910 | 13.3000 |

## Benchmark regression analysis

The benchmark regression model is first used to test the direct effect of technological innovation to enhance the level of green development. The results of Hausman's test indicate that a fixed effect model should be chosen. The results of Hausman's test indicate that a fixed effect model should be chosen. Table 3 exhibits the benchmark regression results of technological innovation and green development. Among them, column (1) presents the regression result with all control variables included. In order to test whether there is a U-shaped relationship between technological innovation and green development, the quadratic terms of core explanatory variables are added to column (1), and the regression results are shown in column (2). As shown in Table 3 column (1), technological innovation has a significant positive promotion effect on green development, so Hypothesis 1 is verified. Meanwhile, the results of Column (2) show that, after introducing the quadratic terms of the core explanatory variables, neither the coefficients of the core explanatory variables nor their quadratic terms are statistically significant. This confirms that there is no U-shaped relationship between technological innovation and green development.

The temporal division into pre- establishment phase and post-establishment phase of the National Sustainable Development Agenda Innovation Demonstration Zone allows for separate regression analyses, with the results presented in columns (3) and (4). Following the decision to establish the National Sustainable Development Agenda Innovation Demonstration Zone in Chengde City, the influence of technological innovation on green development transitions from a negative impact to a notably positive one. This observation suggests that the implementation of the National Sustainable Development Agenda Innovation Demonstration Zone policy can substantially amplify the positive catalytic effect of technological innovation on promoting green development.

As far as the control variables are concerned, it's important to note that there is a non-significant negative correlation between the scale of fiscal expenditure (FE) and population density (PD) and green development. This indicates that neither the scale of fiscal expenditure (FE) nor population density (PD) significantly contributes to the improvement of the level of the region's green development. Furthermore, the coefficients of the level of urbanization (UR) and the scale of fiscal revenue (FR) are positive but non-significant, indicating that an

**Table 3. Benchmark regression results of technological innovation affecting green development.**

| Variable | (1) | (2) | (3) | (4) |
|---|---|---|---|---|
|  | GD | GD | GD | GD |
| TI | 0.0101**(0.0036) | -0.0135(0.0182) | -0.0023(0.0060) | 0.0317**(0.0100) |
| TI$^2$ | — | 0.0022(0.0014) |  |  |
| UR5 | 0.0007(0.0025) | 0.0007(0.0024) | 0.0005(0.0030) | 0.0050(0.0031) |
| FE12 | -0.0011(0.0027) | -0.0007(0.0025) | 0.0015(0.0031) | -0.0027(0.0023) |
| PD11 | -0.0029(0.0054) | -0.0035(0.0062) | 0.0008(0.0066) | -18.1819**(7.6515) |
| FR7 | 0.0032(0.0021) | 0.0047*(0.0025) | 0.0027(0.0036) | 0.0204***(0.0042) |
| _cons | 0.2321(0.1914) | -0.2263(0.1904) | 0.1464(0.1844) | 89.3118**(37.5119) |
| County FE | YES | YES | YES | YES |
| Year FE | YES | YES | YES | YES |
| Observation | 77 | 77 | 77 | 77 |
| R$^2$ | 0.9405 | 0.9420 | 0.9740 | 0.9740 |

Note: *, **, ***, respectively, mean significance at the level of 10%, 5%, and 1%, the number in brackets is the t value, which is calculated by a county-level clustering robust standard error, the same below.

increase in the urbanization rate and the scale of fiscal revenue does not increase the level of green development at the same time. The coefficients of urbanization level (UR) and fiscal revenue scale (FR) are positive but not significant, indicating that the increase of urbanization rate and fiscal revenue scale does not effectively promote green development level.

## Endogeneity and robustness tests

Although the fixed effects model and the inclusion of control variables have been used to deal with the endogeneity problem due to omitted variables, there may still be endogeneity problems due to bidirectional causation, etc., resulting in non-consistent and biased estimated coefficients. The increase in technological innovation leads to more green development, but higher levels of green development, in turn, encourage more technological innovation, i.e., the null hypothesis is affected by the endogeneity of bidirectional causation. Referring to Zhang et al. [39], we adopt lagged one-period technological innovation as an instrumental variable. This choice is grounded in the fact that technological innovation possesses a certain historical coherence, with the level of technological innovation in the previous period influencing the current period's technological innovation. Furthermore, after controlling for relevant demographic variables, economic factors, and incorporating city and year fixed effects, it is established that technological innovation in the previous period does not directly impact the change in green development in the current period. Instead, its influence is mediated solely through the technological innovation of the current period, making it unrelated to the disturbance term of the current period.

The instrumental variable regression results are shown in Table 4. From the results in column (2) of Table 4, it can be seen that, even after introducing the instrumental variables, technological innovation still shows a significant positive impact on green development. Furthermore, the regression coefficient of technological innovation is larger than that in the baseline regression model, which indicates that the instrumental variable could effectively address the endogeneity problem. The conclusion that technological innovation promotes significantly the improvement of the level of green development is still solidly established after solving the problem. In addition, the F-value of the instrumental variable in the first stage is 13.043, which is greater than 10, and also indicates that the selection of the instrumental variable is valid.

To ensure the reliability of the research findings, we further adopt the following methods for validation: first, one lag is applied to the core explanatory variables; second, one lag is applied to the control variables; third, one lag is applied to both the core explanatory variables and the control variables; and fourth, replacing the model. Considering the impact of restricted dependent variables, we use Tobit model to replace benchmark regression model and re-run

**Table 4. Results of endogeneity and robustness tests.**

| Variables | GD | | | | |
|---|---|---|---|---|---|
| | 2SLS (1) | Lagged TI (2) | Lagged control (3) | All lags (4) | Tobit model (5) |
| TI | 0.0550*(0.0271) | 0.0117**(0.0040) | 0.0221**(0.0078) | 0.0131***(0.0022) | 0.0303***(0.0088) |
| _cons | — | 0.2961(0.2353) | 0.2863(0.1833) | 0.2192(0.2105) | -0.3707**(0.1453) |
| Control | YES | YES | YES | YES | YES |
| County FE | YES | YES | YES | YES | — |
| Year FE | YES | YES | YES | YES | — |
| Observation | 77 | 77 | 77 | 77 | 77 |

the regression analysis. From the robustness results, the coefficients of technological innovation are all significantly positive at the 5% level, which fully indicates that the benchmark regression results are robust.

## Threshold effect test

Previous research has established that technological innovation has a significant promotion effect on green development. However, does the promotion effect of technological innovation on green development vary depending on the level of R&D investments and the level of industrial structure upgrading? In other words, is there an appropriate interval between the level of R&D investments and the level of industrial structure upgrading? Within what range of R&D investments and industrial structure upgrading level is the promotion effect of technological innovation on green development most significant? To answer the above questions, we explore whether there are threshold effects of R&D investments and industrial structure upgrading on the impact of technological innovation on green development by constructing threshold effect models with R&D investments and industrial structure upgrading as threshold variables.

Firstly, a panel threshold existence test is conducted based on the method of Hansen [28]. After the "Bootstrap" repeated sampling 300 times, the single threshold test, double threshold test, and triple threshold test were carried out to determine whether there is a threshold and the number of thresholds, and the results of the self-sampling test of the number of thresholds are shown in Table 5. After the self-sampling test of the number of thresholds, the threshold estimate and its 95% confidence interval can be obtained, and the results are shown in Table 6. In order to determine the authenticity of the threshold values, Fig 5 shows the estimation of the threshold value and the construction process of the confidence intervals. It can be seen that the impact of technological innovation on green development is subject to a single threshold effect of R&D investments at a 10% significance level, with a threshold value of 1.1211, thereby hypothesis 2 is proved. Similarly, the impact of technological innovation on green development is subject to a single threshold of industrial structure upgrading at the 1% significance level, with a threshold value of 2.3145, so hypothesis 3 is proved.

After determining the number of thresholds, threshold estimated values, and confidence intervals, the regression results with R&D investments and industrial structure upgrading as threshold variables are shown in Table 7. Under different levels of R&D investments, there is a significant difference in the degree of influence of technological innovation on green development. When R&D investments falls below the threshold value of 1.1211, technological

**Table 5. Self-sampling test results for the number of thresholds.**

| Threshold number test | Model | F value | P value | Bootstrap count | Critical value | | |
|---|---|---|---|---|---|---|---|
| | | | | | 1% | 5% | 10% |
| Threshold effect of R&D investments on the impact of technological innovation on green development | Single Threshold | 10.26 | 0.0433 | 300 | 14.8275 | 11.6430 | 9.0214 |
| | Double threshold | 9.23 | 0.1167 | 300 | 17.8059 | 12.6446 | 9.6624 |
| | Triple threshold | 7.31 | 0.3300 | 300 | 21.9002 | 16.6628 | 12.9617 |
| Threshold effect of industrial structure upgrading on the impact of technological innovation on green development | Single Threshold | 16.62 | 0.0167 | 300 | 16.9632 | 12.7929 | 9.5996 |
| | Double threshold | 10.20 | 0.1433 | 300 | 18.1729 | 13.3714 | 11.4531 |
| | Triple threshold | 1.92 | 0.9467 | 300 | 32.1632 | 22.0402 | 17.1882 |

**Table 6. Threshold estimates and their confidence intervals.**

| Threshold effect test | Model | Threshold estimate | 95% confidence interval |
|---|---|---|---|
| Threshold effect of R&D investments on the impact of technological innovation on green development | Single Threshold | 1.1211 | [1.0602, 1.1286] |
| Threshold effect of industrial structure upgrading on the impact of technological innovation on green development | Single Threshold | 2.3145 | [2.3132, 2.3360] |

innovation has a significant promotional effect on green development, with a regression coefficient of 0.0241, which passes the 1% significance test. When the R&D investments exceeds the threshold value of 1.1211, technological innovation continues to have a significant role in promoting green development, with a regression coefficient of 0.0405, which also passes the 1% significance test. This indicates that with the continuous increase of R&D investments, the promotion effect of technological innovation on green development shows an increasing trend, thus further proving the judgment of Hypothesis 2. It does not appear that technological innovation has an inhibitory effect on green development due to excessive R&D investment. This can be mainly explained by the fact that the level of R&D investment in Chengde is low and may not have reached the level of the threshold. Therefore, to maximize the positive effect of technological innovation on green development, it is essential to maintain the level of R&D investments within the appropriate interval, specifically exceeding the threshold value of 1.1211.

Under different levels of industrial structure upgrading, there is a significant disparity in the degree of influence of technological innovation on green development. When the level of industrial structure upgrading falls below the threshold value of 2.3145, technological innovation has a significant promoting effect on green development, with a regression coefficient of 0.0178, passing the 10% significance test. Similarly, when the level of industrial structure upgrading surpasses the threshold value of 2.3145, technological innovation also has a significant promoting effect on green development, with a regression coefficient of 0.0212, and passes the 1% significance test. This indicates that with the continuous improvement of industrial structure upgrading level, the promotion effect of technological innovation on green development shows an increasing trend, proving the judgment of Hypothesis 3. Therefore, to

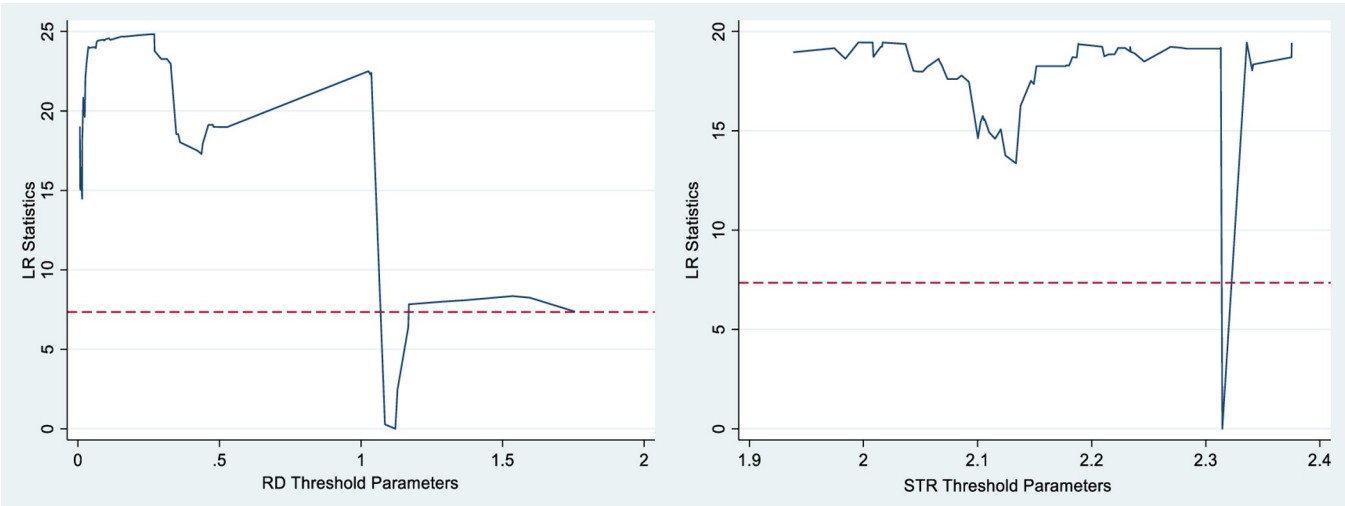

**Fig 5. Single threshold estimated values and confidence intervals.**

**Table 7. Threshold regression results for the impact of technological innovation on green development.**

| Variable | Threshold Variables | |
|---|---|---|
| | (1)RD | (2)STR |
| Threshold value q | 1.1211 | 2.3145 |
| TI*I(Th≤q) | 0.1292[**](0.0046) | 0.0182[***](0.0055) |
| TI*I(Th≥q) | 0.0282[***](0.0047) | 0.0308[***](0.0060) |
| control | YES | YES |
| _cons | -0.3925[**](0.1691) | -0.5187[***](0.1122) |
| Observation | 77 | 77 |

fully leverage the promotional effect of technological innovation on green development, it is crucial to maintain the level of industrial structure upgrading within the appropriate range, specifically exceeding the threshold value of 2.3145.

## Discussion

Numerous scholars have empirically demonstrated the driving effect of technological innovation on green development, yet little attention has been paid to assessing whether this impact varies across different conditions. Utilizing Chengde National Sustainable Development Agenda Innovation Demonstration Zone as a case study, this research integrates the triad of R&D investments, industrial structure, and technological innovation into a comprehensive analytical framework. The objective is to explore whether these elements collectively form the capacity to coordinate development and foster green development. The study concludes that technological innovation indeed exhibits a catalytic effect on green development, aligning with prior scholarly findings [17–22]. Furthermore, it reveals a pronounced driving effect of technological innovation on green development following the implementation of the National Sustainable Development Agenda Innovation Demonstration Zone policy. Consequently, the anticipated "rebound effect" [40] triggered by technological innovation is not observed. Considering the feedback loop of technological innovation on green development—where increased technological innovation leads to more green development, and higher levels of green development, in turn, encourage further technological innovation—the empirical results indicate a reverse causal relationship between technological innovation and green development. To address endogeneity, this paper employs the instrumental variable approach.

Additionally, a singular threshold emerges concerning the impact of technological innovation on green development, determined by the R&D investment levels and industrial structure upgrading levels. The positive effects of technological innovation on green development can only be fully realized when the thresholds are surpassed. It is noteworthy that, in contrast to existing scholarly findings, technological innovation does not appear to exert an inhibitory effect on green development due to excessive R&D investment. This discrepancy is primarily attributed to the relatively low level of R&D investment in Chengde, which may not have reached the critical threshold.

The framework and findings of this paper shed light on the intricate relationship between technological innovation and green development from the perspectives of R&D investment and industrial structure, contributing to the field of research on technological innovation and green development. However, there are some limitations and shortages in this study, as limited below: (1) This study only explores the impact mechanism of technological innovation on green development, and does not explore the impact mechanism of technological innovation on sustainable development. Therefore, in the next stage, we will focus on the relationship

between technological innovation and sustainable development. (2) This study only takes Chengde City, a NSDAIDZ established in 2019, as an example. Therefore, in the next stage, this study will conduct an in-depth study with 13 NSDAIDZs as research objects to obtain more meaningful conclusions and provide more targeted development strategies.

## Conclusion

Empirical testing is conducted using a panel of balanced data from counties, cities, and districts within Chengde City, classified as a NSDAIDZ. The data spans the years 2014 to 2020. This study aims to investigate the impact and threshold effect of technological innovation on green development in NSDAIDZs. Building upon established findings linking technological innovation to green development, we delve deeper into the ramifications of technological advancement within the context of current policies. Utilizing the Chengde National Sustainable Development Agenda Innovation Demonstration Zone as a case study, we try to reveal the impact of technological innovation on green development in this policy context. At the same time, we assess the potential of R&D investment, industrial structure, and technological innovation to synergistically foster development and propel green initiatives, thereby addressing gaps in previous research.

The results show that: (1) Technological innovation has a positive promotion effect on green development at a 5% significance level, i.e., for each unit increase in technological innovation, the level of green development increases by 1.01% in NSDAIDZs; (2) There is a noteworthy disparity in the impact of technological innovation on green development before and after the establishment of National Sustainable Development Agenda Innovation Demonstration Zone; (3) The impact of technological innovation on green development is subject to a single threshold of R&D investments, with the threshold value of 1.1211. The impact of technological innovation on green development is subject to a single threshold of industrial structure upgrading at the 1% significance level, with a threshold value of 2.3145. Threshold effect tests reveal that only when the level of R&D investment and industrial structure upgrading cross a certain threshold, the positive effect of technological innovation on green development can be fully released.

### Policy implications

The establishment of NSDAIDZs is the major initiative of China's Country Program for the Implementation of the 2030 Agenda for Sustainable Development, and also provides experience and reference for other countries and regions in the world with similar sustainable development problems. The Chengde NSDAIDZ brings opportunities and challenges for green development, innovation, and sustainable development. Promoting green development in the NSDAIDZs through technological innovation has become imperative. Synthesizing the full-text research, we argue:

1. Reinforce the impact of policy support, bolster the capacity for technological innovation, and ignite the vibrancy of innovation. Aligned with the directives of the National Sustainable Development Agenda Innovation Demonstration Zone, we will rigorously execute national and provincial science and technology innovation policies. On one hand, we aim to steer enterprises towards escalating R&D investments, thereby stimulating their innovation capabilities. On the other hand, we seek to encourage social capital to actively engage in and support corporate research initiatives. This multifaceted approach aims to comprehensively facilitate the interconnected development of science, technology, industry, and finance.

2. Attach importance to the role of technological innovation in environmental governance, improving the quality of growth, and ecological protection. In terms of environmental governance and ecological protection, technological innovation is used to provide green products, processes, etc., which reduce resource consumption and environmental pollution and improve environmental governance capacity and ecological protection efficiency. In terms of economic growth, on one hand, through strengthening technological innovation to improve production efficiency, reduce production costs, and improve economic efficiency. On the other hand, through technological innovation to accelerate the greening of the production process and the process of renewable and recyclable, promote the transformation and upgrading of the high-input, high-consumption, high-pollution, crude production model, as well as build a green, low-carbon, sustainable production system.

3. Proactively nurture the positive interaction among R&D investment, industrial structure, technological innovation, and green development. Thoughtfully elevating the level of R&D investment and optimizing industrial structure can synergistically complement sustainable development support policies, thereby reinforcing the role of technological innovation in advancing green development.

## Supporting information

**S1 Data.**
(DTA)

## Acknowledgments

We would like to thank members of our team, for their helpful feedback in drafting previous versions of this manuscript.

## Author Contributions

**Conceptualization:** Guofeng Zhang, Jiangao Niu.

**Data curation:** Qingqing Yuan, Ruixian Wang, Xiaojing Ma.

**Formal analysis:** Qingqing Yuan, Ruixian Wang.

**Methodology:** Guofeng Zhang.

**Writing – original draft:** Qingqing Yuan.

**Writing – review & editing:** Guofeng Zhang.

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
