## [Decision Letter · Decision Letter 0]

7 Dec 2023

PONE-D-23-37873Does Technological Innovation in National Sustainable Development Agenda Innovation Demonstration Zones Promote Green Development? --The Case from Chengde City, ChinaPLOS ONE

Dear Dr. Zhang,

Thank you for submitting your manuscript to PLOS ONE. After careful consideration, we feel that it has merit but does not fully meet PLOS ONE’s publication criteria as it currently stands. Therefore, we invite you to submit a revised version of the manuscript that addresses the points raised during the review process.

We look forward to receiving your revised manuscript.

Kind regards,

Fuyou Guo, (Ph.D.

Academic Editor

PLOS ONE

Journal Requirements:

   "This study was supported by the Key Public Project of Humanity and Social Sciences Research of Hebei Education Department (No. ZD202311), General Project of Soft Science Research of Hebei Provincial Department of Science and Technology (No.225576117D), Graduate Student Innovation Funding Project of Hebei Province (No. CXZZSS2023137)"

    "the Key Public Project of Humanity and Social Sciences Research of Hebei Education Department (No. ZD202311); General Project of Soft Science Research of Hebei Provincial Department of Science and Technology (No.225576117D); Graduate Student Innovation Funding Project of Hebei Province (No. CXZZSS2023137)"

   "the Key Public Project of Humanity and Social Sciences Research of Hebei Education Department (No. ZD202311); General Project of Soft Science Research of Hebei Provincial Department of Science and Technology (No.225576117D); Graduate Student Innovation Funding Project of Hebei Province (No. CXZZSS2023137)"  

6. Please ensure that you refer to Figure 1 in your text as, if accepted, production will need this reference to link the reader to the figure.

Additional Editor Comments:

Reviewer 1 Comments:

1. The introduction section is too long and needs to be presented more in terms of scholarly contributions.

2. The chapter numbering in the whole text needs to be carefully proofread, especially in Mechanism and research hypotheses.

3. Compared with Eq. (1), TI in Eq. (2) is not subscripted, and the squared term is incorrectly written as X.

4. The selection of indicators needs to be clarified and a valid literature base needs to be added.

5. The selection of instrumental variables needs to be explained appropriately.

6. The empirical results need to be compared with similar studies.

7. the conclusion section should be supplemented with the implications of this paper in terms of academic research

Reviewer 2 Comments:

Some issues that should be seriously considered before acceptance:

1. Regression Model Specification:

Authors should clearly explain why square terms are introduced and elaborate on their theoretical basis in addressing the research question. The use of square terms should align with the theoretical framework in the research area. Otherwise, besides simply adding square terms, consider other more flexible nonlinear modeling methods such as spline regression, nonparametric estimation, or polynomial regression. Additionally, the use of square terms introduces challenges in interpreting nonlinear effects. The authors should provide more detailed explanations to ensure readers understand how square terms affect the dependent variable.

2. Endogeneity Issues:

I am concerned about the endogeneity issues in this article, such as

a) Omitted Variables:

Problem: e.g. The model may not have considered the impact of urban planning policies, which could be a crucial omitted variable.

Explanation: If urban planning policies affect both technological innovation and urban green development and are not considered in the model, the model might underestimate the actual impact of technological innovation on urban development.

b) Bidirectional Causation:

Problem: The model may not have considered the feedback effects of technological innovation on green development.

Explanation: If an increase in technological innovation leads to more green development, and higher levels of green development, in turn, encourage more technological innovation, the model might underestimate the true relationship between technological innovation and green development. In this case, the model might need to more comprehensively consider this mutual influence to accurately capture the causation between the two.

3. Instrumental Variable Issues:

If the lagged one-period variables used as instrumental variables are endogenous (i.e., influenced by common unobserved factors), the instrumental variable may still be subject to endogeneity issues, rendering it ineffective in addressing endogeneity. Additionally, lagged variables may be unstable, especially in the presence of external shocks or structural changes. This may cause the instrumental variable's effects to vary over time, making it ineffective during different time periods, and you assume structural changes indeed exist. Besides the instrumental variable method, have you considered other methods to address endogeneity? Please discuss the reasons for choosing the instrumental variable method and, where possible, consider comparing it with other endogeneity handling methods.

4. Threshold Effects:

In the manuscript, there needs to be a clearer expression of the theoretical basis for threshold effects. Explain why the study assumes the existence of threshold effects and their rationale in addressing the research question. Although threshold values are mentioned in the text, there seems to be no detailed discussion on how this threshold value is chosen. In threshold effect studies, the choice of the threshold is crucial as it directly influences the interpretation of study results. A thorough theoretical and empirical argumentation is needed for the selection of the threshold value, and consideration should be given to whether there are multiple potential threshold values. When explaining threshold effects, ensure that the results are interpretable to readers. Clearly state the significance of threshold effects for the research question and why threshold effects make theoretical sense. While statistical significance is important, also pay attention to substantive significance. In threshold effect studies, ensure that the threshold effects are not only statistically significant but also substantively meaningful in practice.

Reviewer 3 Comments:

The authors take Chengde City as the case study to investigate the impact of technological innovation on green development, additionally, taking the Research and development investments as the threshold variable. Overall, the topic is interesting. However, there are some issues, from my point of view, deserve to be treated before publication.

1.I would suggest a bit rework of the paper in the introduction. In its current version, the paper spends too much ink on the description of national policies, but the elaboration of the relationship between technological innovation, green development, R&D investment, and industrial structure upgrading is a bit abrupt and lacks corresponding references, which needs to be further improved.

2.The research framework, including R&D investments, industrial structure, technological innovation and green development, needs to be further clarified. Technological innovation and R&D investment are very broad concepts that involve various industries, and probably only some of them are related to green development. Therefore, I am wondering whether it is possible to change these variables to be more relevant to green development.

3. The authors argue that the work is novel because it uses Chengde City, the NSDAIDZs, as a case study to investigate the impact of technological innovation on green development, which has previously been overlooked. I have two questions about this. First, Chengde City was identified as the NSDAIDZs in May 2019, but the panel data in this paper is selected from 2014 to 2020. It would be better if the study period could be extended to compare the changes in this effect before and after the establishment of the NSDAIDZs. Second, the theme of Chengde being identified as the NSDAIDZs is closely related to its water resources, and these can be discussed. Third, is it possible to have a map to illustrate the location of Chengde in China?

Reviewer 4 Comments:

This article takes Chengde city as an example to examine the quantitative impact of technological innovation on green development in the National Sustainable Development Agenda Innovation Demonstration Zones (NSDAIDZs), and the topic has certain practical significance. However, the article has the following obvious problems: (1) The innovation is not obvious, and there is already a lot of research on the impact of technological innovation on green development, and the difference between this article and existing research is not significant. (2) The conclusion of the article can only indicate that technological innovation in Chengde has promoted green development, but cannot indicate that this impact is caused by NSDAIDZs policy. In fact, the empirical research in this article cannot exclude the influence of other policies or factors. (3) The theoretical mechanism is too generalized. Even without NSDAIDZs policies, technological innovation can still promote green development, and the conclusion is obvious. The article lacks in-depth analysis of the specific implementation content and characteristics of China's NSDAIDZs policy. (4) The article only uses data from 11 districts in Chengde City from 2014 to 2020, with a small sample size. The empirical research methods and robustness tests are too simple, and the basis is insufficient, which will affect the reliability of the conclusions. (5) The conclusions obtained have no effect on how to further improve NSDAIDZs, and the targeted and actionable policy recommendations are not strong.

Reviewers' comments:

Reviewer's Responses to Questions

**Comments to the Author**

1. Is the manuscript technically sound, and do the data support the conclusions?

Reviewer #1: Yes

Reviewer #2: Yes

Reviewer #3: Yes

Reviewer #4: No

2. Has the statistical analysis been performed appropriately and rigorously? 

Reviewer #1: Yes

Reviewer #2: Yes

Reviewer #3: Yes

Reviewer #4: No

3. Have the authors made all data underlying the findings in their manuscript fully available?

Reviewer #1: Yes

Reviewer #2: Yes

Reviewer #3: Yes

Reviewer #4: No

4. Is the manuscript presented in an intelligible fashion and written in standard English?

Reviewer #1: Yes

Reviewer #2: Yes

Reviewer #3: Yes

Reviewer #4: No

5. Review Comments to the Author

Reviewer #1: 1. The introduction section is too long and needs to be presented more in terms of scholarly contributions.

2. The chapter numbering in the whole text needs to be carefully proofread, especially in Mechanism and research hypotheses.

3. Compared with Eq. (1), TI in Eq. (2) is not subscripted, and the squared term is incorrectly written as X.

4. The selection of indicators needs to be clarified and a valid literature base needs to be added.

5. The selection of instrumental variables needs to be explained appropriately.

6. The empirical results need to be compared with similar studies.

7. the conclusion section should be supplemented with the implications of this paper in terms of academic research

Reviewer #2: Some issues that should be seriously considered before acceptance:

1. Regression Model Specification:

Authors should clearly explain why square terms are introduced and elaborate on their theoretical basis in addressing the research question. The use of square terms should align with the theoretical framework in the research area. Otherwise, besides simply adding square terms, consider other more flexible nonlinear modeling methods such as spline regression, nonparametric estimation, or polynomial regression. Additionally, the use of square terms introduces challenges in interpreting nonlinear effects. The authors should provide more detailed explanations to ensure readers understand how square terms affect the dependent variable.

2. Endogeneity Issues:

I am concerned about the endogeneity issues in this article, such as

a) Omitted Variables:

Problem: e.g. The model may not have considered the impact of urban planning policies, which could be a crucial omitted variable.

Explanation: If urban planning policies affect both technological innovation and urban green development and are not considered in the model, the model might underestimate the actual impact of technological innovation on urban development.

b) Bidirectional Causation:

Problem: The model may not have considered the feedback effects of technological innovation on green development.

Explanation: If an increase in technological innovation leads to more green development, and higher levels of green development, in turn, encourage more technological innovation, the model might underestimate the true relationship between technological innovation and green development. In this case, the model might need to more comprehensively consider this mutual influence to accurately capture the causation between the two.

3. Instrumental Variable Issues:

If the lagged one-period variables used as instrumental variables are endogenous (i.e., influenced by common unobserved factors), the instrumental variable may still be subject to endogeneity issues, rendering it ineffective in addressing endogeneity. Additionally, lagged variables may be unstable, especially in the presence of external shocks or structural changes. This may cause the instrumental variable's effects to vary over time, making it ineffective during different time periods, and you assume structural changes indeed exist. Besides the instrumental variable method, have you considered other methods to address endogeneity? Please discuss the reasons for choosing the instrumental variable method and, where possible, consider comparing it with other endogeneity handling methods.

4. Threshold Effects:

In the manuscript, there needs to be a clearer expression of the theoretical basis for threshold effects. Explain why the study assumes the existence of threshold effects and their rationale in addressing the research question. Although threshold values are mentioned in the text, there seems to be no detailed discussion on how this threshold value is chosen. In threshold effect studies, the choice of the threshold is crucial as it directly influences the interpretation of study results. A thorough theoretical and empirical argumentation is needed for the selection of the threshold value, and consideration should be given to whether there are multiple potential threshold values. When explaining threshold effects, ensure that the results are interpretable to readers. Clearly state the significance of threshold effects for the research question and why threshold effects make theoretical sense. While statistical significance is important, also pay attention to substantive significance. In threshold effect studies, ensure that the threshold effects are not only statistically significant but also substantively meaningful in practice.

Reviewer #3: The authors take Chengde City as the case study to investigate the impact of technological innovation on green development, additionally, taking the Research and development investments as the threshold variable. Overall, the topic is interesting. However, there are some issues, from my point of view, deserve to be treated before publication.

1.I would suggest a bit rework of the paper in the introduction. In its current version, the paper spends too much ink on the description of national policies, but the elaboration of the relationship between technological innovation, green development, R&D investment, and industrial structure upgrading is a bit abrupt and lacks corresponding references, which needs to be further improved.

2.The research framework, including R&D investments, industrial structure, technological innovation and green development, needs to be further clarified. Technological innovation and R&D investment are very broad concepts that involve various industries, and probably only some of them are related to green development. Therefore, I am wondering whether it is possible to change these variables to be more relevant to green development.

3. The authors argue that the work is novel because it uses Chengde City, the NSDAIDZs, as a case study to investigate the impact of technological innovation on green development, which has previously been overlooked. I have two questions about this. First, Chengde City was identified as the NSDAIDZs in May 2019, but the panel data in this paper is selected from 2014 to 2020. It would be better if the study period could be extended to compare the changes in this effect before and after the establishment of the NSDAIDZs. Second, the theme of Chengde being identified as the NSDAIDZs is closely related to its water resources, and these can be discussed. Third, is it possible to have a map to illustrate the location of Chengde in China?

Reviewer #4: This article takes Chengde city as an example to examine the quantitative impact of technological innovation on green development in the National Sustainable Development Agenda Innovation Demonstration Zones (NSDAIDZs), and the topic has certain practical significance. However, the article has the following obvious problems: (1) The innovation is not obvious, and there is already a lot of research on the impact of technological innovation on green development, and the difference between this article and existing research is not significant. (2) The conclusion of the article can only indicate that technological innovation in Chengde has promoted green development, but cannot indicate that this impact is caused by NSDAIDZs policy. In fact, the empirical research in this article cannot exclude the influence of other policies or factors. (3) The theoretical mechanism is too generalized. Even without NSDAIDZs policies, technological innovation can still promote green development, and the conclusion is obvious. The article lacks in-depth analysis of the specific implementation content and characteristics of China's NSDAIDZs policy. (4) The article only uses data from 11 districts in Chengde City from 2014 to 2020, with a small sample size. The empirical research methods and robustness tests are too simple, and the basis is insufficient, which will affect the reliability of the conclusions. (5) The conclusions obtained have no effect on how to further improve NSDAIDZs, and the targeted and actionable policy recommendations are not strong.

6. PLOS authors have the option to publish the peer review history of their article (what does this mean?). If published, this will include your full peer review and any attached files.

Reviewer #1: No

Reviewer #2: **Yes: **Yifu Yang

Reviewer #3: No

Reviewer #4: No

---

## [Author Response · Author response to Decision Letter 0]

3 Feb 2024

Report of the First Reviewer -- PONE-D-23-37873 /Zhang

Comments

1. The introduction section is too long and needs to be presented more in terms of scholarly contributions.

Response: 

We thank the reviewer's comments and suggestions. We agree with the reviewer that introduction section is too long and needs to be presented more in terms of scholarly contributions. We have revised the related discussions. 

Changes to the manuscript: 

Following the reviewer’s suggestion, we have adjusted the structure of the introduction section and we have included the discussion about scholarly contributions. [Page 8, Line 157-165]

2. The chapter numbering in the whole text needs to be carefully proofread, especially in Mechanism and research hypotheses.

Response: 

We thank the reviewer's suggestions. We agree with the reviewer that the chapter numbering in the whole text needs to be carefully proofread. We have revised the related chapter numbers.

Changes to the manuscript: 

We have adjusted the chapter number in Mechanism and research hypotheses. [Page 8, Line 171; Page 9, Line 197-198; Page 11, Line 228-229]

3. Compared with Eq. (1), TI in Eq. (2) is not subscripted, and the squared term is incorrectly written as X.

Response: 

We thank the reviewer's suggestions. We agree with the reviewer that Eq. (2) does have the writing error. And now Eq. (2) has been modified to Eq. (4).

Changes to the manuscript: 

Following the reviewer’s suggestion, we have corrected Eq. (2). Now Eq. (2) has been modified to Eq. (4). [Page 13, Line 267]

4. The selection of indicators needs to be clarified and a valid literature base needs to be added.

Response: 

We appreciate the reviewers' suggestion. We've added the reasons and process for selecting the indicators. At the same time, literature has been added as a support. In addition, considering that the number of R&D personnel and the amount of R&D investment have a certain relationship with population density and local government fiscal revenue and expenditure, which may affect the research results, this paper uses the latter to measure the level of R&D investment.

Changes to the manuscript: 

We have added the literature base to the selection of core explanatory and threshold variables. [Page 16, Line 327-331; Page 17, Line 339; Page 17, Line 340; Page 17, Line 342]

5. The selection of instrumental variables needs to be explained appropriately.

Response: 

We appreciate the reviewers' suggestion. Following the reviewer’s suggestion, we have included the explanation for the selection of instrumental variables. In line with the fundamental principles of constructing instrumental variables, a suitable instrument must satisfy two key conditions: first, it should exhibit a robust correlation with the endogenous variables, and second, it must be exogenous and independent of the error term. Referring to Zhang et al[1]., we adopt lagged one-period technological innovation as an instrumental variable, a common practice in the existing literature. This choice is grounded in the fact that technological innovation possesses a certain historical coherence, with the level of technological innovation in the previous period influencing the current period's technological innovation. Furthermore, after controlling for relevant demographic variables, economic factors, and incorporating city and year fixed effects, it is established that technological innovation in the previous period does not directly impact the change in green development in the current period. Instead, its influence is mediated solely through the technological innovation of the current period, making it unrelated to the disturbance term of the current period. Consequently, the selected indicator logically fulfills both criteria for instrumental variables.

Changes to the manuscript: 

In the revised version, with the help of helpful suggestions from reviewers, we have included the explanation for the selection of instrumental variables. [Page 22, Line 430-439]

6. The empirical results need to be compared with similar studies.

Response: 

We appreciate the reviewer for the efforts to improve the quality of our manuscripts. We agree and accept the reviewer's comments. In the discussion section, we compare the empirical results with similar studies, and supplement the implications and shortcomings of this paper in terms of academic research.

Changes to the manuscript: 

In the revised version, with the help of helpful suggestions from reviewers, we have added the comparison in the part of discussion. [Page 27, Line 526-552]

7. the conclusion section should be supplemented with the implications of this paper in terms of academic research

Response: 

We appreciate the reviewers' suggestion. Following the reviewer’s suggestion, we have included the discussion about the implications of this paper in terms of academic research. 

Changes to the manuscript: 

Building upon established findings linking technological innovation to green development, this study delves deeper into the ramifications of technological advancement within the context of current policies. Utilizing the Chengde National Sustainable Development Agenda Innovation Demonstration Zone as a case study, it seeks to discern whether the impact of technological innovation varies under distinct conditions. The paper assesses the potential of R&D investment, industrial structure, and technological innovation to synergistically foster development and propel green initiatives, thereby addressing gaps in previous research. [Page 29, Line 570-578]

Report of the Second Reviewer -- PONE-D-23-37873 /Zhang

Comments

 Regression Model Specification: Authors should clearly explain why square terms are introduced and elaborate on their theoretical basis in addressing the research question. The use of square terms should align with the theoretical framework in the research area. Otherwise, besides simply adding square terms, consider other more flexible nonlinear modeling methods such as spline regression, nonparametric estimation, or polynomial regression. Additionally, the use of square terms introduces challenges in interpreting nonlinear effects. The authors should provide more detailed explanations to ensure readers understand how square terms affect the dependent variable.

Response: 

We appreciate the reviewers' suggestion about the regression model specification. We have added a discussion on whether there is a U-shaped relationship between technological innovation and green development in the mechanism study, and added quadratic terms of core explanatory variables to the empirical study for testing. Because the quadratic term of the core explanatory variable in the fixed-effect model did not pass the test, we did not proceed with the U-test to prove the authenticity of the U-shaped relationship.

We couldn't agree more strongly that in addition to simply adding square terms, there are other, more flexible nonlinear modeling methods to consider, such as spline regression, nonparametric estimation, or polynomial regression. However, based on the previous research on whether the relationship between technological innovation and green development is a U-shaped relationship, this paper chooses to add a square term.

Changes to the manuscript: 

“However, scholars continue to debate whether the relationship between technological innovation and green development is linear. It is not universally true that higher levels of technological innovation always result in more significant promotion of green development. When the level of technological innovation is low, the efficiency of resource and energy utilization is low; along with the enhancement of technological innovation, advanced cleaner and other production technologies have been widely used, promoting the reduction of resource and energy consumption and emissions of three wastes. Therefore, there may be a U-shaped relationship between technological innovation and green development.” [Page 9, Line 185-193]

“Among them, column (1) presents the regression result with all control variables included. In order to test whether there is a U-shaped relationship between technological innovation and green development, the quadratic terms of core explanatory variables are added to column (1), and the regression results are shown in column (2).” [Page 19, Line 386-390]

Endogeneity Issues: I am concerned about the endogeneity issues in this article, such as 

a) Omitted Variables: Problem: e.g. The model may not have considered the impact of urban planning policies, which could be a crucial omitted variable. Explanation: If urban planning policies affect both technological innovation and urban green development and are not considered in the model, the model might underestimate the actual impact of technological innovation on urban development.

Response: 

We thank the reviewer’s comments. We agree and accept the reviewer's comments. There are two major endogenous problems in identifying the impact of technological innovation on green development. The first is the problem of missing variables. In the benchmark regression model, multiple factors such as the scale of fiscal expenditure, population density and urbanization rate that affect green development should be controlled as much as possible, and the fixed effects of regions and years should be controlled to reduce the endogeneity caused by the omission of variables. However, technological innovation may still be affected by other uncontrolled third-party factors or policies, which may also affect green development. As the reviewers mentioned, the model may not have considered the impact of urban planning policies, which could be a crucial omitted variable. 

In order to promote sustainable development, Chengde has formulated the Chengde Sustainable Development Plan (2018-2030) (Revised in 2022) and the Chengde National Sustainable Development Agenda Innovation Demonstration Zone Construction Plan (2021-2025) in recent years. It has played an important role in the construction of the demonstration area. However, due to the difficulty of obtaining data on urban planning policies in the county data used in this paper, the impact of urban planning policies on technological innovation and green development is not considered.

b) Bidirectional Causation:

Problem: The model may not have considered the feedback effects of technological innovation on green development.

Explanation: If an increase in technological innovation leads to more green development, and higher levels of green development, in turn, encourage more technological innovation, the model might underestimate the true relationship between technological innovation and green development. In this case, the model might need to more comprehensively consider this mutual influence to accurately capture the causation between the two.

Response: 

We appreciate the reviewer for the efforts to improve the quality of our manuscripts. We agree and accept the reviewer's comments. It is true that the model does not take into account the feedback effect of technological innovation on green development. If an increase in technological innovation leads to more green development, and higher levels of green development, in turn, encourage more technological innovation. Therefore, we examine the endogeneity by examining the impact of green development on technological innovation.

Changes to the manuscript: 

“Considering the feedback effects of technological innovation on green development, i.e., if an increase in technological innovation leads to more green development, and higher levels of green development, in turn, encourage more technological innovation. This study examines the endogeneity by examining the impact of green development on technological innovation. The results show that green development has a significant positive impact on technological innovation.” [Page 21, Line 425-431]

 Instrumental Variable Issues:

If the lagged one-period variables used as instrumental variables are endogenous (i.e., influenced by common unobserved factors), the instrumental variable may still be subject to endogeneity issues, rendering it ineffective in addressing endogeneity. Additionally, lagged variables may be unstable, especially in the presence of external shocks or structural changes. This may cause the instrumental variable's effects to vary over time, making it ineffective during different time periods, and you assume structural changes indeed exist. Besides the instrumental variable method, have you considered other methods to address endogeneity? Please discuss the reasons for choosing the instrumental variable method and, where possible, consider comparing it with other endogeneity handling methods.

Response: 

We appreciate the reviewers' suggestion. Following the reviewer’s suggestion, we have included the explanation for the selection of instrumental variables. In line with the fundamental principles of constructing instrumental variables, a suitable instrument must satisfy two key conditions: first, it should exhibit a robust correlation with the endogenous variables, and second, it must be exogenous and independent of the error term. Referring to Zhang et al., we adopt lagged one-period technological innovation as an instrumental variable, a common practice in the existing literature. This choice is grounded in the fact that technological innovation possesses a certain historical coherence, with the level of technological innovation in the previous period influencing the current period's technological innovation. Furthermore, after controlling for relevant demographic variables, economic factors, and incorporating city and year fixed effects, it is established that technological innovation in the previous period does not directly impact the change in green development in the current period. Instead, its influence is mediated solely through the technological innovation of the current period, making it unrelated to the disturbance term of the current period. Consequently, the selected indicator logically fulfills both criteria for instrumental variables.

Besides the instrumental variable method, we have considered other methods to address endogeneity, such as DID-GMM model and SYS-GMM model. The SYS-GMM model stands out as a superior estimation method, proficiently addressing issues such as autocorrelation, heteroskedasticity, endogeneity, and weak instrumental variables. This approach ensures the production of unbiased, efficient, and consistent estimation results. Notably, the SYS-GMM model adeptly tackles the challenge of endogeneity between the explained variables and their lagged counterparts, mitigating the risks of biased and inconsistent model analysis. In examining the impact of technological innovation on regional green development, we opt for the SYS-GMM model. Here, regional green development (GD) serves as the explanatory variable, incorporating the lagged 1-period green development into the model. The core explanatory variable is technological innovation (TI). This model selection meets the condition of having a dynamic explanatory variable and not all explanatory variables strictly being exogenous. Moreover, it includes controls for individual and time double fixed effects. The results of the SYS-GMM model regression are shown in the table below.

Table. SYS-GMM test

Variables GMM teat

TI 0.0499**(0.0225)

Control YES

Observation 77

AR (1) 0.042

AR (2) 0.137

Hansen 1.000

The significance of the core explanatory variables remained consistent, and the p-values for both the Hansen and AR(2) test statistics exceeded 10%. This lack of statistical significance prevents the rejection of the original hypothesis, indicating the absence of autocorrelation. Consequently, the rationale behind the establishment of the SYS-GMM model is justified, and the study's findings are validated as robust.

Both the system GMM model and the instrumental variables method are pivotal techniques for addressing endogeneity issues. However, the system GMM model exhibits several drawbacks in comparison to the instrumental variables method: (1) the necessity to choose the lag order in the system GMM model, with improper selection potentially introducing uncertainty into the re

---

## [Decision Letter · Decision Letter 1]

27 Feb 2024

Does Technological Innovation in National Sustainable Development Agenda Innovation Demonstration Zones Promote Green Development? --The Case from Chengde City, China

PONE-D-23-37873R1

Dear Dr. Zhang,

We’re pleased to inform you that your manuscript has been judged scientifically suitable for publication and will be formally accepted for publication once it meets all outstanding technical requirements.

Kind regards,

Fuyou Guo, (Ph.D.

Academic Editor

PLOS ONE

Additional Editor Comments (optional):

Reviewers' comments:

Reviewer's Responses to Questions

**Comments to the Author**

1. If the authors have adequately addressed your comments raised in a previous round of review and you feel that this manuscript is now acceptable for publication, you may indicate that here to bypass the “Comments to the Author” section, enter your conflict of interest statement in the “Confidential to Editor” section, and submit your "Accept" recommendation.

Reviewer #1: All comments have been addressed

Reviewer #4: All comments have been addressed

2. Is the manuscript technically sound, and do the data support the conclusions?

Reviewer #1: Yes

Reviewer #4: Yes

3. Has the statistical analysis been performed appropriately and rigorously? 

Reviewer #1: Yes

Reviewer #4: Yes

4. Have the authors made all data underlying the findings in their manuscript fully available?

Reviewer #1: (No Response)

Reviewer #4: Yes

5. Is the manuscript presented in an intelligible fashion and written in standard English?

Reviewer #1: Yes

Reviewer #4: Yes

6. Review Comments to the Author

Reviewer #1: (No Response)

Reviewer #4: (No Response)

7. PLOS authors have the option to publish the peer review history of their article (what does this mean?). If published, this will include your full peer review and any attached files.

Reviewer #1: No

Reviewer #4: No

---

## [Editor Report · Acceptance letter]

26 Apr 2024

PONE-D-23-37873R1 

PLOS ONE

Dear Dr. Zhang, 

I'm pleased to inform you that your manuscript has been deemed suitable for publication in PLOS ONE. Congratulations! Your manuscript is now being handed over to our production team.

Kind regards, 

on behalf of

Associate professor Fuyou Guo 

Academic Editor

PLOS ONE